# Dual-Resolution Correspondence Networks

**Xinghui Li**[1], **Kai Han**[2], **Shuda Li**[1], and **Victor Prisacariu**[1]

[1]Active Vision Lab, University of Oxford
[1]{xinghui, shuda, victor}@robots.ox.ac.uk
[2]Visual Geometry Group, University of Oxford
[2]khan@robots.ox.ac.uk

## Abstract

We tackle the problem of establishing dense pixel-wise correspondences between a pair of images. In this work, we introduce Dual-Resolution Correspondence Networks (DualRC-Net), to obtain pixel-wise correspondences in a coarse-to-fine manner. DualRC-Net extracts both coarse- and fine- resolution feature maps. The coarse maps are used to produce a full but coarse 4D correlation tensor, which is then refined by a learnable neighbourhood consensus module. The fine-resolution feature maps are used to obtain the final dense correspondences guided by the refined coarse 4D correlation tensor. The selected coarse-resolution matching scores allow the fine-resolution features to focus only on a limited number of possible matches with high confidence. In this way, DualRC-Net dramatically increases matching reliability and localisation accuracy, while avoiding to apply the expensive 4D convolution kernels on fine-resolution feature maps. We comprehensively evaluate our method on large-scale public benchmarks including HPatches, InLoc, and Aachen Day-Night. It achieves the state-of-the-art results on all of them.

## 1   Introduction

Establishing correspondences between a pair of images that shares a common field of view is critical for many important tasks in computer vision such as 3D reconstruction [1, 2, 3], camera motion estimation [4], relocalization [5, 6, 7, 8], etc. Most existing methods tackle the problem using a classic 3-stage pipeline, i.e., detection, description and matching [7, 9]. They first detect local repeatable salient points known as the keypoints, then describe these keypoints by extracting feature vectors from some local regions around them, and finally form the keypoint pairs, also known as the correspondences, by selecting a set of high confident matches from all possible candidate matches. The resulting correspondences can then be used to triangulate 3D coordinates for model reconstruction [1] or to feed the RANSAC Perspective-$n$-Point (P$n$P) algorithms [10, 11] for camera pose estimation [12, 13]. This pipeline has demonstrated great success in dealing with certain illumination or camera viewpoint variations using the classic hand-crafted local feature descriptors such as SIFT [14], SURF [15], BRISK [16] or the modern CNN features [17]. However, many of these methods are based on local support regions and therefore suffer from the missing detection problems [6]. During matching, they only consider up to $k$ nearest neighbours, instead of the entire matching space, which also limits their performance especially under extreme appearance changes such as day-to-night with wide viewpoint variations.

Recently, several approaches [18, 19, 5] aim to avoid the detection stage by considering every point from a regular grid for matching. As a result, dense matches can be obtained by retrieving the confident ones from all possible candidate matches. Hence, the missing detection problem can be alleviated. Among these approaches, the Neighbourhood Consensus Networks (NCNet) [5] and its variants [20, 21, 6] have shown encouraging results. These methods employ a CNN to extract features

from two images, then calculate a 4D correlation tensor representing the entire matching space where each location records the cosine correlation score between a pair of feature vectors, and refine the correlation tensor by a sequence of 4D convolution kernels. The learnt 4D kernels turn out to be highly effective in filtering incorrect matches. However, the use of 4D convolution kernels in these methods are prohibitively expensive, restricting them from handling images with a high resolution or resulting in inadequate localisation accuracy.

In this paper, we propose Dual-Resolution Correspondence Networks (DualRC-Net) to establish dense correspondences for high-resolution images in a coarse-to-fine manner. DualRC-Net first extracts both coarse- and fine-resolution feature maps. Specifically, we employ an FPN-like [22] feature backbone to obtain the dual-resolution feature maps. Then, the coarse maps are used to produce a full 4D correlation tensor, which is then refined by the neighbourhood consensus module. The fine-resolution feature maps are used to obtain the final dense correspondences guided by the refined coarse 4D correlation tensor. Specifically, the full 4D tensor is used to select local regions in the high-resolution feature maps where the corresponding regions have high correlation scores, avoiding calculating the expensive full 4D correlation tensor for the high-resolution feature maps. When both the coarse and fine correlation scores are coupled together, more reliable correspondences can be estimated from the candidate local regions for the high-resolution feature maps. This can drastically reduce the memory footprint and computation cost while boosting the matching results for more accurate localization.

The main contributions of this paper can be summarised as follow: First, we introduce a novel neural network architecture which generates dual-resolution feature maps allowing to match correspondences in a coarse-to-fine manner; second, the rough matches extracted from the coarse correlation tensor allow the model to focus on the local regions in the fine-resolution feature maps that are very likely to contain the correct matches. This dramatically reduces the memory footprint and computation cost for matching on fine-resolution feature maps; third, we comprehensively evaluate our method on large-scale public benchmarks including HPatches [23], InLoc [24], and Aachen Day-Night [13] achieving the state-of-the-art results on all of them, which demonstrates the effectiveness as well as generalisability of the proposed method. Our code can be found at `https://code.active.vision`.

## 2 Related work

Establishing correspondences between two images has been investigated extensively for decades [25, 26, 19, 18], and it is the workhorse for many computer vision tasks such as 3D reconstruction, image retrieval, robot relocalization, etc. It is out of the scope of the paper to review all methods for correspondence estimation, and here we only review the most relevant ones.

The most popular approaches follow the 3-stage pipeline of detection, description and matching. Detecting salient and repeatable keypoints [27, 28, 29, 30, 7] allows a small memory and computation cost by the detected sparse 2D points, which can then be described using either hand-crafted feature descriptors [14, 15, 16, 31] or data-driven learning-based approaches [32, 33, 7, 8, 34, 35, 36, 37]. Finally, these keypoints with feature descriptors can then be paired together to form a candidate matching space from which a set of confident matches can be retrieved using random sampling schemes [38, 39] or validity check by inter-feature constraints [40, 3]. Although popular, the 3-stage pipeline suffers the missing detection problem [6]. In addition, each stage was trained separately, which brings extra complexity to deploy.

Recently, NCNet [5] was introduced to combine all the three stages into an end-to-end trainable framework while effectively incorporating the neighbourhood consensus constraint to filter outliers. In particular, NCNet first constructs a dense 4D matching tensor that contains all the possible matches in the feature space, and each element in the 4D matching tensor represents the matching score of one possible match. In this way, all possible pairs are considered, therefore, this could remedy the missing detection problem to some extent. NCNet then applies a sequence of 4D convolutions on the 4D matching tensor to realize the neighbour consensus constraint to refine the matching results. It has shown promising results for correspondence estimation. Inspired by NCNet, several methods have been proposed to improve this framework by using self-similarity to capture complex local pattern for matching [20, 21] and introducing non-isotropic 4D filtering to better deal with scale variation [21]. One main drawback of NCNet is large memory consumption. The size of the 4D matching score tensor increases quadratically with the size of the images. Therefore, NCNet is not scalable to images

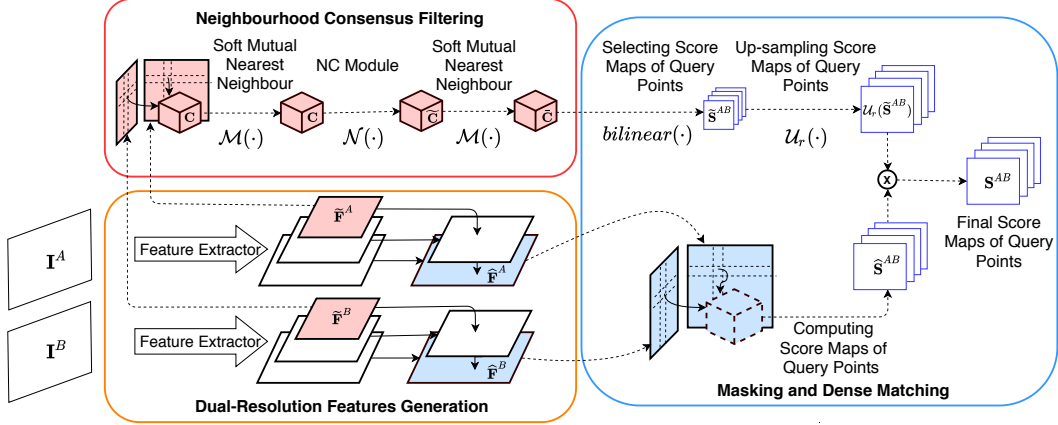

Figure 1: Overview of DualRC-Net. The coarse feature map $\widetilde{\mathbf{F}}^A$ and $\widetilde{\mathbf{F}}^B$ (red) are combined to form the 4D correlation tensor $\mathbf{C}$ which is then refined by the neighbourhood consensus module. The refined 4D tensor $\bar{\mathbf{C}}$ can be used to generate the 2D score map $\widetilde{\mathbf{S}}^{AB}$. After up-sampling $\widetilde{\mathbf{S}}^{AB}$, it can be used to selected confident local regions in the fine-resolution feature maps $\widehat{\mathbf{F}}^A$ and $\widehat{\mathbf{F}}^B$ (blue) and to adjust the 2D correlation score map $\widehat{\mathbf{S}}^{AB}$ from the fine-resolution feature maps to get the final correlation score map $\mathbf{S}^{AB}$, from which the dense correspondence can be retrieved. Best viewed in colour.

with a high resolution. Very recently, we notice the concurrent work Sparse-NCNet [6] that tackles the problem by projecting the 4D correlation map into a 3D sub-manifold where ordinary convolution can be applied to achieve similar or better performance to NCNet. However, projecting the original 4D correlation to the sub-manifold might lose some useful information for matching. In contrast, our method maintains dual-resolution feature maps. We subtly use the coarse-resolution feature map to form a refined 4D matching tensor, which is used to adjust the matching scores obtained using the fine-resolution features. In this way, our method can enjoy both the benefits of robust trainable neighbourhood consensus constraint and denser matching from the fine-resolution feature maps. As will be shown, our method can obtain better results than Sparse-NCNet using a smaller input image size.

Another series of relevant works are the CNN based methods for camera pose estimation [41, 42, 43, 44], where a pair of images are fed into a Siamese network [41, 42] to directly regress the relative pose. These methods are simple and effective, but typically applied to deal with small viewpoint changes and suffers limited capability of generalising to new scenes.

## 3 Method

We tackle the dense correspondence estimation problem by introducing the Dual-Resolution Correspondence Networks (DualRC-Net) which establish correspondences in a coarse-to-fine manner. Figure 1 provides an overview of the DualRC-Net. Specifically, we employ an FPN-like feature extractor to build both coarse- and fine-resolution feature maps. The coarse-resolution feature maps are then used to form the 4D correlation tensor which is then refined by the learnable neighbourhood consensus module. The refined 4D correlation tensor is then used to select local regions in the high-resolution feature maps where the corresponding regions in the refined 4D correlation tensor have high correlation scores. The final dense matches are then obtained by combining the local correlation maps from fine-resolution features with their corresponding 2D correlation maps generated from the refined 4D correlation tensor.

In the remainder of the section, we will first briefly describe the neighbourhood consensus module [35, 5, 20] in section 3.1, then describe our strategy to generate the dual-resolution feature maps in section 3.2, and then introduce our method using the dual-resolution feature maps for high accuracy dense correspondence estimation in section 3.3. The training loss is described in section 3.4.

## 3.1 Neighbourhood consensus filtering

The neighbourhood consensus using 4D convolutions was initially introduced in [5]. Given a pair of images $\mathbf{I}^A$ and $\mathbf{I}^B$, their feature maps $\mathbf{F}^A \in \mathbb{R}^{C \times H_a \times W_a}$ and $\mathbf{F}^B \in \mathbb{R}^{C \times H_b \times W_b}$ can be extracted using a standard CNN feature backbone (e.g., VGG16 [45], ResNet [46]), where $H$ and $W$ are the height and width of the feature map and $C$ is the number of channels. The complete matching space for all possible pairs of locations can be represented by a 4D tensor $\mathbf{C} \in \mathbb{R}^{H_a \times W_a \times H_b \times W_b}$ with each element being the cosine correlation $\mathbf{C}_{ijkl} = \mathbf{f}_{ij}^{A\top} \mathbf{f}_{kl}^B / (\|\mathbf{f}_{ij}^A\|_2 \|\mathbf{f}_{kl}^B\|_2)$ where $\mathbf{f}_{ij}^I \in \mathbb{R}^C$ represents the feature vector at location $(i, j)$ in $\mathbf{F}^I$, and $\|.\|_2$ represents the $L$-2 norm. Essentially, $\mathbf{C}$ contains all possible combinations of feature pairs in $\mathbf{F}^A$ and $\mathbf{F}^B$. A local 4D neighbourhood in $\mathbf{C}$ actually represents the matching consistency between two local regions from the images. If a match is correct, their neighbourhood consistency should be relatively high. Therefore, 4D convolutions can be applied to learn the consistency patterns from the training data. If we denote the 4D convolution layers as a function $\mathcal{N}(.)$, the neighbourhood consensus filtering can be represented as $\widetilde{\mathbf{C}} = \mathcal{N}(\mathbf{C}) + \mathcal{N}(\mathbf{C}^\top)^\top$ where $\top$ denotes the matching direction swapping of two images, i.e., $\mathbf{C}_{ijkl}^\top = \mathbf{C}_{klij}$. This is to avoid the learning to be biased in one matching direction.

In addition, a cyclic matching constraint can be employed in the form of soft mutual nearest neighbour filter $\bar{\mathbf{C}} = \mathcal{M}(\widetilde{\mathbf{C}})$ such that any resulting correlation score $\bar{\mathbf{C}}_{ijkl} \in \bar{\mathbf{C}}$ is high only if both $\widetilde{\mathbf{C}}_{ijkl}$ and $\widetilde{\mathbf{C}}_{klij}$ are high. Particularly, $\bar{\mathbf{C}}_{ijkl} = r_{ijkl}^A r_{ijkl}^B \widetilde{\mathbf{C}}_{ijkl}$, where $r_{ijkl}^A = \widetilde{\mathbf{C}}_{ijkl} / \max_{ab} \widetilde{\mathbf{C}}_{abkl}$ and $r_{ijkl}^B = \widetilde{\mathbf{C}}_{ijkl} / \max_{cd} \widetilde{\mathbf{C}}_{ijcd}$. We apply the filtering before and after the neighbourhood consensus module following [5].

## 3.2 Dual-resolution feature map generation

It has been reported in the literature [5, 47, 6] that the resolution of the feature maps to constitute the 4D correlation tensor affects the accuracy. However, the high memory and computation cost of 4D tensors prevents most existing works from scaling to large feature maps. To avoid calculating the complete 4D tensor of the high-resolution feature maps, we propose to use the dual-resolution feature maps (see Figure 2), from which we can enjoy the benefits of both 4D convolution based neighbourhood consensus on coarse-resolution feature maps and more reliable matching from the fine-resolution feature maps.

In particular, we adopt a FPN-like [22] feature backbone, which can extract a pair of interlocked coarse-resolution feature map $\widetilde{\mathbf{F}} \in \mathbb{R}^{h \times w}$ and fine-resolution feature map $\widehat{\mathbf{F}} \in \mathbb{R}^{H \times W}$. The intuition of using FPN is that it fuses the contextual information from the top layers in the feature hierarchy into the bottom layers that mostly encode low-level information. In such way, the bottom layers can not only preserve their resolution but also contain rich high-level information, which makes them more robust for matching. Particularly, to maintain the high descriptiveness of the main feature backbone, we increase the number of output channels of the $1 \times 1$ conv kernels from 256 to that of the highest feature map (i.e. 1024). The aim is to increase the complexity of the descriptor and hence achieve higher accuracy during matching. The resolution for $\widehat{\mathbf{F}}$ is 4 times of $\widetilde{\mathbf{F}}$.

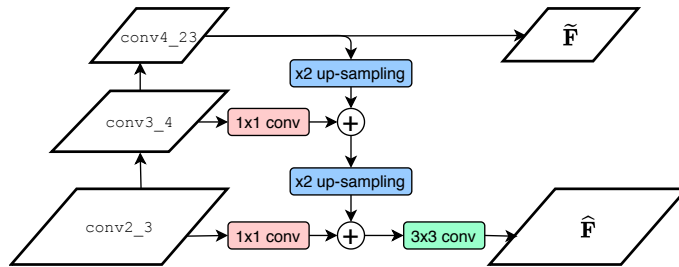

Figure 2: Dual-resolution feature map extractor. The fine-resolution feature map $\widehat{\mathbf{F}}$ is 4 times larger than the coarse-resolution feature map $\widetilde{\mathbf{F}}$, allowing denser, more reliable and better localised matches.

Such a dual-resolution structure allows us to establish a progressive matching pipeline. The coarse-resolution feature map $\widetilde{\mathbf{F}}$ firstly produces 4D correlation tensor which is subsequently filtered by the

neighbourhood consensus module. The refined 4D score tensor is then used to identify and re-weight the local regions in the fine-resolution feature map $\widehat{\mathbf{F}}$, from which the final dense correspondences are obtained.

## 3.3 Dense matching

Given a feature vector $\hat{\mathbf{f}}_{ij}^A \in \widehat{\mathbf{F}}^A$, we compute its cosine correlation with all features in the target feature map $\widehat{\mathbf{F}}^B$ and generate a 2D score map $\widehat{\mathbf{S}}_{ij}^{AB} \in \mathbb{R}^{H_b \times W_b}$ by

$$\widehat{\mathbf{S}}_{ij,kl}^{AB} = \frac{\langle \mathbf{f}_{ij}^A, \mathbf{f}_{kl}^B \rangle}{\left\| \mathbf{f}_{ij}^A \right\|_2 \left\| \mathbf{f}_{kl}^B \right\|_2}, \quad \text{for all } k \in \{0, 1, ..., H_b - 1\} \text{ and all } l \in \{0, 1, ..., W_b - 1\}. \quad (1)$$

Let $r = \frac{H_a}{h_a} = \frac{W_a}{w_a}$ be the ratio between spatial dimensions of fine- and coarse-resolution feature maps, and $(i', j')$ be the corresponding location of $(i, j)$ on coarse-resolution feature map $\widetilde{\mathbf{F}}^A$. We have $\frac{i}{i'} = \frac{j}{j'} = r$. Notice that $(i', j')$ may not be integers and hence we apply bilinear interpolation $bilinear(\cdot)$ to extract the 2D score map $\widetilde{\mathbf{S}}_{i'j'}^{AB}$ from $\bar{\mathbf{C}}$ by

$$\widetilde{\mathbf{S}}_{i'j',k'l'}^{AB} = bilinear(\bar{\mathbf{C}}_{\lceil i' \rceil \lceil j' \rceil k'l'}, \bar{\mathbf{C}}_{\lceil i' \rceil \lfloor j' \rfloor k'l'}, \bar{\mathbf{C}}_{\lfloor i' \rfloor \lceil j' \rceil k'l'}, \bar{\mathbf{C}}_{\lfloor i' \rfloor \lfloor j' \rfloor k'l'}), \quad (2)$$

where $\lceil \cdot \rceil$ and $\lfloor \cdot \rfloor$ denote the ceiling and floor of the input respectively, $\widetilde{\mathbf{S}}_{i'j'}^{AB} \in \mathbb{R}^{h_b \times w_b}$, $k' \in \{0, 1, ..., h_b - 1\}$ and $l' \in \{0, 1, ..., w_b - 1\}$. This 2D score map $\widetilde{\mathbf{S}}_{i'j'}^{AB}$ will then serve as a "mask" to adjust the fine-resolution score map $\widehat{\mathbf{S}}_{ij}^{AB}$. For each score on $\widetilde{\mathbf{S}}_{i'j'}^{AB}$, it corresponds to an $r \times r$ region on $\widehat{\mathbf{S}}_{ij}^{AB}$. Hence we up-sample $\widetilde{\mathbf{S}}_{i'j'}^{AB}$ by $r$ times to the same size as $\widehat{\mathbf{S}}_{ij}^{AB}$ using the nearest neighbour. Let $\mathcal{U}_r(\cdot)$ be this up-sampling operation, we can then use $\widetilde{\mathbf{S}}_{i'j'}^{AB}$ to mask over $\widehat{\mathbf{S}}_{ij}^{AB}$ to produce the final score map $\mathbf{S}_{ij}^{AB}$ by

$$\mathbf{S}_{ij}^{AB} = \widehat{\mathbf{S}}_{ij}^{AB} \odot \mathcal{U}_r(\widetilde{\mathbf{S}}_{i'j'}^{AB}), \quad (3)$$

where $\odot$ represents the element-wise multiplication between two matrices.

For point $(i, j)$ in $\widehat{\mathbf{F}}^A$, its correspondence in $\widehat{\mathbf{F}}^B$ is then retrieved by $(k^*, l^*) = \operatorname{argmax}_{kl} \mathbf{S}_{ij,kl}^{AB}$. By querying all $(i, j)$ on $\widehat{\mathbf{F}}^A$, we can obtain a set of dense matches $\{(i_n, j_n), (k_n, l_n)\}_{n=1}^{N}$. In order to filter the outliers, we adopt the mutual nearest neighbour criteria. Namely, we obtain the dense match sets from both $\widehat{\mathbf{F}}^A$ to $\widehat{\mathbf{F}}^B$ and $\widehat{\mathbf{F}}^B$ to $\widehat{\mathbf{F}}^A$, and only keep the intersetion of two sets as our final set of matches. However, exhaustively querying all features from the fine-resolution feature map is time-consuming. Hence, we adopt a simple strategy to speed up the matching procedure. Firstly, we establish a set of matches for all features in coarse feature map $\widetilde{\mathbf{F}}^A$, denoted as $\{(i'_n, j'_n), (k'_n, l'_n)\}_{n=1}^{M}$, which can be directly retrieved from $\bar{\mathbf{C}}$. We then sort scores of these matches in descending order and select top $50\%$ of them. As each spatial location on $\widetilde{\mathbf{F}}^A$ corresponds to an $r \times r$ region in $\widehat{\mathbf{F}}^A$, we query all features in $\widehat{\mathbf{F}}^A$ that lie in the corresponding local region of top $50\%$ of $\{(i'_n, j'_n)\}_{n=1}^{M}$ and apply the mutual neighbour criteria as described above. This effectively reduces the number of queries by half and increases speed.

## 3.4 Training loss

Due to the absence of dense keypoint annotations, existing methods normally use either image-level pairwise annotations [5, 20, 6] or sparse keypoint annotations [21] for training. We follow [21] to adopt the training loss based on sparse keypoint annotations. Specifically, the loss is defined as

$$\mathcal{L}_k = \|\mathbf{M}^{AB} - \mathbf{M}_{gt}^{AB}\|_F + \|\mathbf{M}^{BA} - \mathbf{M}_{gt}^{BA}\|_F, \quad (4)$$

where $\mathbf{M}^{AB} \in \mathbb{R}^{N \times T}$ is the probability map given $N$ query keypoints in the source image w.r.t. the whole frame of the target feature map $\widehat{\mathbf{F}}^B$. $T = H_b \times W_b$ is the total number of elements in the fine-resolution feature map. Each row in $\mathbf{M}^{AB}$ represents the vectorized score map $\mathbf{S}_{ij}^{AB}$ for a given query point $(i, j)$ in the source image. $\mathbf{M}_{gt}^{AB}$ is the ground truth. $\|.\|_F$ denotes the Frobenius norm. To provide smoother learning signals from the ground truth, we follow [21] to apply the Gaussian

blurring on $\mathbf{M}_{gt}^{AB}$ and $\mathbf{M}_{gt}^{BA}$. We also apply the orthogonal loss proposed by [21] and use it as a regularization term to enforce one-to-one match. It is defined as

$$\mathcal{L}_o = \|\mathbf{M}^{AB}\mathbf{M}^{AB^\top} - \mathbf{M}_{gt}^{AB}\mathbf{M}_{gt}^{AB^\top}\|_F + \|\mathbf{M}^{BA}\mathbf{M}^{BA^\top} - \mathbf{M}_{gt}^{BA}\mathbf{M}_{gt}^{BA^\top}\|_F. \qquad (5)$$

Overall, the loss can be written as

$$\mathcal{L} = \mathcal{L}_k + \lambda\mathcal{L}_o, \qquad (6)$$

where $\lambda$ is a weight term, which is set to $0.05$ in our experiments.

**Implementation details**  We implemented our pipeline in Pytorch [48]. For feature extractor, we use ResNet101 pre-trained on ImageNet and truncate the part after `conv4_23`. It is fixed during training. For fine-resolution feature maps, we extract the output of `conv2_3`, `conv3_4` and fuse the output of `conv4_23` and `conv3_4` into the output of `conv2_3` in the way depicted in Figure 2. The $1 \times 1$ conv layers and the $3 \times 3$ conv fusing layers are fine-tuned. Comparison on different variants of the dual-resolution feature backbone can be found in the supplementary. For the neighbourhood consensus module, we use the same configuration as [5]. We train our model using Adam optimizer [49] for 15 epochs with an initial learning rate of $0.01$ which is halved every 5 epochs.

**Training data**  Following [7], we train our models on MegaDepth dataset [50], which consists of a large number of internet images about 196 scenes and their sparse 3D point clouds are constructed by COLMAP [51, 52]. The camera intrinsic and extrinsic together with the depth maps of $102, 681$ are also included. We also follow [7] to generate sparse ground truth labels. First, we compare the overlap in sparse SfM point cloud between all image pairs to select the pairs whose overlap is over $50\%$. Next, for all selected pairs, the second image with depth information is projected into the first image and occlusion is removed by depth check. Then, we randomly collect $128$ correspondences from each image pair to train our model. We use the scenes with more than $500$ valid image pairs for training and the rest scenes for validation. To avoid scene bias, $110$ image pairs are randomly selected from each training scene to constitute our training set. In total, we obtain $15, 070$ training pairs and $14, 638$ validation pairs. After training, we evaluate our model on HPatches, InLoc and Aachen Day-Night to validate the effectiveness and generalisability.

### 3.5  HPatches

HPatches benchmark [23] contains homography patches under significant illumination and viewpoint change. We follow the evaluation protocol of [6], where 108 image sequences are evaluated, with 56 of them about viewpoint change and 52 of them about illumination change. Each sequence consists of one reference image and five query images. Each query image pairs with one reference image hence five image pairs are obtained for each sequence. Homography w.r.t. to the reference image is provided for each query image. For each image pairs, matches in query image are projected into reference image using homography provided. We adopt the commonly used Mean Matching Accuracy (MMA) as evaluation metric

$$\text{MMA}(\{\mathbf{p}_i^A, \mathbf{p}_i^B\}_{i=1}^N; t) = \frac{\sum_{i=1}^N \mathbb{1}(t - \|H(\mathbf{p}_i^A) - \mathbf{p}_i^B\|_2)}{N}, \qquad (7)$$

where $t$ is the threshold of 2D distance. $\mathbb{1}(\cdot)$ is a binary indicator function whose output is 1 for non-negative value and 0 otherwise. $H(\cdot)$ denotes the warping by homography.

**DualRC-Net vs other neighbourhood consensus methods**  In Figure 3, we compare our approach with two neighbourhood consensus based methods, namely the NCNet [5] baseline, and the very recent Sparse-NCNet [6] which shows the state-of-the-art result. As can be seen, our method outperforms both Sparse-NCNet and NCNet by a significant margin under illumination change. In general, our method performs on par with Sparse-NCNet under viewpoint change. Sparse-NCNet slightly outperforms our method with a threshold smaller than 6 pixels, while our method outperforms Sparse-NCNet with a threshold larger than 6 pixels. Overall, our method notably outperforms both competitor methods. The feature map resolution is the key for robust matching. To have the feature map with a resolution of $200 \times 150$, NCNet needs to take an image with a resolution of $3200 \times 2400$. Sparse-NCNet can obtain a feature map resolution of $400 \times 300$ with the same input image size by reducing the stride of the last convolution layer in the feature extractor. In contrast, our method can obtain a feature map with the resolution of $400 \times 300$ by only using the input image with a resolution of $1600 \times 1200$. Figure 4 demonstrates the qualitative results of two image pairs. More comparison and qualitative results can be found in the supplementary.

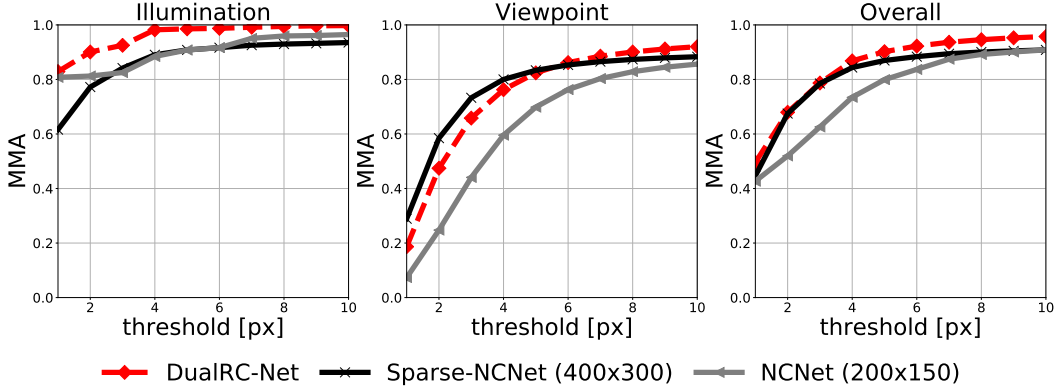

Figure 3: DualRC-Net vs other neighbourhood consensus based methods on HPatches. For each method, top 2000 matches are selected for evaluation.

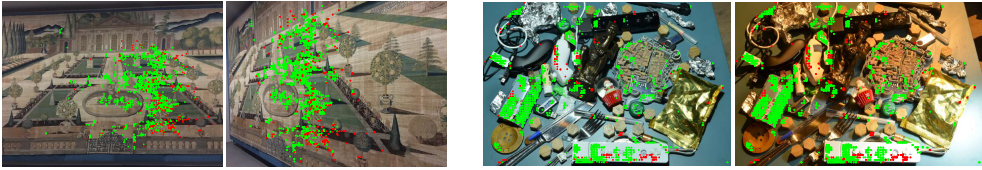

Figure 4: Qualitative results on HPatches. Top 2000 matches are selected with $t = 3$ pixels. Our method is robust to huge viewpoint (left pair) and illumination (right pair) changes. Green and red dots denote correct and incorrect matches respectively (best viewed in PDF with zoom).

**DualRC-Net vs other state-of-the-art methods**    In Figure 5, we compare our method with D2-Net [7], DEFL [53], R2D2 [8], SuperPoint [33] and SuperPoint+SuperGlue [40]. The general trend is similar to the comparison with NCNet and Sparse-NCNet. Overall, DualRC-Net substantially outperforms all other methods.

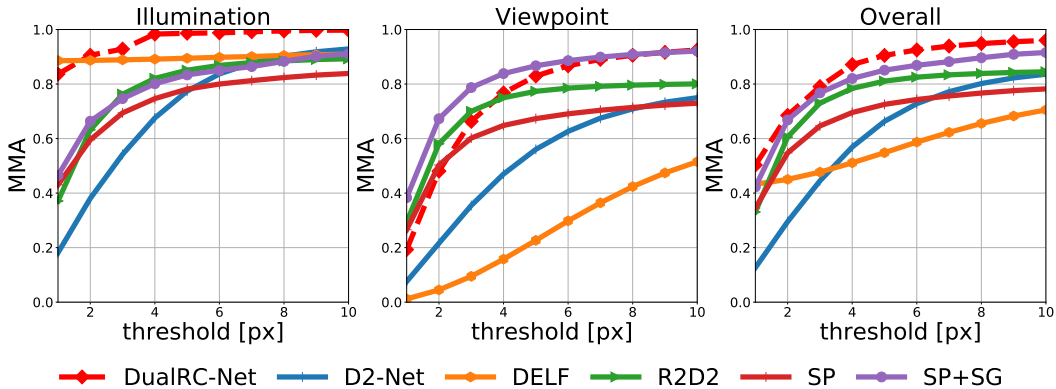

Figure 5: DualRC-Net vs other state-of-the-art methods on HPatches. For other methods, top 2000 feature points are selected. By enforcing mutual nearest neighbour, they roughly generate 1000 matches for each image pairs. Hence we select top 1000 matches for fair comparison.

## 3.6 InLoc

InLoc benchmark [24] is designed to test the performance on long-term indoor relocalization. It consists of a large collection of images with depth acquired by the 3D scanner. Query images are taken by a smartphone camera a few months later. The benchmark contains very challenging viewpoint and illumination changes. We follow the evaluation protocol in [24]. For each query image, 10 candidate database images are selected. DualRC-Net then establishes the correspondence between each pairs.

Finally PnP solver is used to estimate the pose of the query image. The result is shown in Figure 6. We achieve the state-of-the-art result with distance threshold > 0.3m and outperform the second best method by a significant margin. More qualitative results can be found in the supplementary.

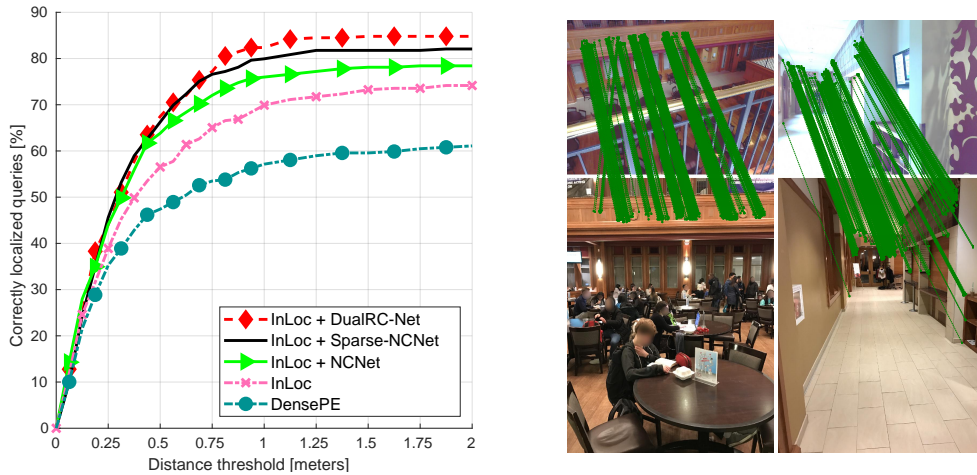

Figure 6: Evaluation on InLoc benchmark. Left: The result on InLoc benchmark. Right: Qualitative results of top 500 matches (best viewed in PDF with zoom). It is clear that our model is able to generate reliable correspondences under large viewpoint and illumination change.

## 3.7 Aachen Day-Night

Aachen Day-Night benchmark [54] is adopted to validate the performance on outdoor relocalization under illumination change. It has 98 night query images and each of them has 20 day-time candidate database images. We use DualRC-Net to establish the correspondences for pose estimation. The result is presented in Figure 7. Our method achieves the state-of-the-art result for thresholds (0.5m, $2°$) and (5m, $10°$). More qualitative results can be found in supplementary.

| Method | Correctly localized queries (%) | | |
|---|---|---|---|
| | 0.5m, $2°$ | 1m, $5°$ | 5m, $10°$ |
| ASLFeat+OANet [9, 3] | 77.6 | 89.8 | **100.0** |
| SP+SG [33, 40] | **79.6** | **90.8** | 100.0 |
| D2-Net [7] | 74.5 | 86.7 | **100.0** |
| R2D2 (K=20k) [8] | 76.5 | **90.8** | 100.0 |
| Sparse-NCNet [6] | 76.5 | 84.7 | 98.0 |
| DualRC-Net | **79.6** | 88.8 | **100.0** |

Figure 7: Evaluation on Aachen Day-Night benchmark. Left: Comparison with other methods. Right: Qualitative results of top 500 matches (best viewed in PDF with zoom).

## 4 Conclusion

We have presented an end-to-end dual-resolution architecture, called DualRC-Net, that can establish dense pixel-wise correspondences between a pair of images in a coarse-to-fine manner. DualRC-Net first obtains a full 4D correlation tensor from the coarse-resolution feature maps, which is refined by the learnable neighbourhood consensus module. The refined correlation tensor is then used to guide the model to obtain robust matching from fine-resolution feature maps. In this way, DualRC-Net dramatically increases matching reliability and localisation accuracy, while avoiding to apply the expensive 4D convolution kernels on fine-resolution feature maps. We comprehensively evaluate our method on large-scale public benchmarks HPatches, InLoc and Aachen Day-Night, achieving the state-of-the-art results.

## Broader Impact

The proposed model enjoys great potential to improve a wide range of industrial applications including image alignment, image retrieval, 3D reconstruction, camera pose estimation, etc. Particularly, the proposed method sets a new record of accuracy on both indoor and outdoor relocalization benchmarks which strongly indicates that it will directly benefit many fields in the near future including robotics, autonomous driving and gaming industry, where the vision-based relocalisation is the foundation. It is particularly important if the application has to work in a GPS-denied environment. Furthermore, with the potential of being able to deploy on mobile platforms such as robots, drones, smartphones, head-mounted displays, the proposed method could be used to boost mixed/virtual reality for entertainment or education. It may be applied to build large-scale long-term indoor/outdoor maps which allows pinpointing a user's location without GPS using images. This may also improve existing navigation capability of robots or drones and enable a more intelligent agent for various tasks such as delivery, searching and rescue.

In the meanwhile, the proposed model, like many other machine learning technology, does have some unwelcomed repercussions. The accurate and robust vision-only correspondence estimation and the subsequent localisation can be used to illegally locate a person or property without permission using an image circulated online. It may even be weaponized to guide a UAV to carry out a terrorism attack. However, these negative impacts are more related to the fields of application rather than the technology itself. Generally, proper legislation may be required to prevent any machine learning method from being used for evil purposes. Fortunately, many countries have already started to debate and evaluate the pros and cons of specific AI technologies. For example, some countries have banned using facial recognition or restricted employing AI for surveillance. Therefore, we believe, under strict supervision, that our work will bring more benefits than harms to society.

## Acknowledgments

We gratefully acknowledge the support of the European Commission Project Multiple-actOrs Virtual EmpathicCARegiver for the Elder (MoveCare) and the EPSRC Programme Grant Seebibyte EP/M013774/1. We are also grateful for the generous help of Mihai Dusmanu and Ignacio Rocco on certain technical issues.

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
