[Supplementary Material]

# Dual-Resolution Correspondence Networks
# –Supplementary Material–

**Xinghui Li**[1], **Kai Han**[2], **Shuda Li**[1], and **Victor Prisacariu**[1]

[1]Active Vision Lab, University of Oxford
[1]{xinghui, shuda, victor}@robots.ox.ac.uk
[2]Visual Geometry Group, University of Oxford
[2]khan@robots.ox.ac.uk

In the supplementary, we present more experimental results and analysis to show the effectiveness of DualRC-Net. In section 1, we provide five alternatives to the FPN-like structure for fusing the dual-resolution feature maps of the feature backbone. In section 2, we compare DualRC-Net with other neighbourhood consensus based methods in more details. Finally, in section 3, we qualitatively compare DualRC-Net with the state-of-the-art methods on three benchmarks. DualRC-Net establishes the new state-of-the-art.

## 1 Investigation on more variants of FPN structure

Apart from the dual-resolution feature extractor we present in the main paper, we also investigate other possible FPN-like architectures (shown in Figure 1) and thoroughly evaluate their effects on the matching performance.

Figure 1: Five variants of FPN architecture. (a) is our default architecture used in the main paper. In (b), we directly fuse the output of `conv4_23` with that of `conv2_3` by up-sampling 4 times. In (c) and (d), additional $3 \times 3$ conv modules are adopted before up-sampling. In (e), we incorporate the the output of `conv5_3`. The channels of all feature maps are aligned to $1024$ by $1 \times 1$ conv layers.

We experiment with different alternatives of the FPN based dual-resolution feature extractor in DualRC-Net and evaluate the model on HPatches [1] as well as InLoc [2] benchmarks. The results are reported in Figure 2 and Figure 3 respectively.

Figure 2: Performance of different variants of FPN architecture on HPatches benchmark. The input image size is $1600 \times 1200$ and the top 1000 matches are selected for the evaluation of MMA.

Figure 3: Performance of different variants of FPN architecture on InLoc benchmark. Input images have the image size of $1600 \times 1200$.

As shown in Figure 2, all five types of variants have similar overall performance. Type (b) and Type (e) are slightly inferior on illumination change at small thresholds but all five types have almost identical performance on viewpoint change. Overall, type (d) very slightly outperforms others but at the cost of more parameters in the $3 \times 3$ convolutional layers than type (a). Similarly, in Figure 3, we can see the five types of variants also have similar performance on InLoc benchmark. Type (b) is marginally behind others while type (a), (c), (d) and (e) have entangled curves such that no type consistently outperform others. These results reveal that type (b) is marginally inferior than other types as it only contains the information from two layers while the rest of the types have almost the same performance. Therefore, we select type (a) which has, except type (b), the simplest architecture among five options as our dual-resolution feature extractor.

Additionally, we also compare type (a) and type (e) with their 256 channel counterparts in Figure 4. Notice that the 256 channel counterpart of type (e) is the original FPN [3]. We can see that increasing number of channels does not affect the performance of type (e). However, type (a) with 1024 channels performs better than that with 256 channels under huge illumination variations at smaller thresholds. This further justifies that type (a) is a more proper choice for DualRC-Net.

Figure 4: Comparison between 256 and 1024 output feature channels for type (a) and type (e). Notice that type (e) with 256 channels is the original FPN architecture.

## 2 Comparison with other neighbourhood consensus based methods

In this section, we compare DualRC-Net with Sparse-NCNet [4] and NCNet [5] on HPatches benchmark in more details. In Figure 5a, we show the performance of the three methods given input images with the same size of $1600 \times 1200$. DualRC-Net achieves the best results in all cases except under huge illumination changes with a threshold less than 3 pixels. The success attributes to that DualRC-Net can generate a larger relocalisation feature map under the same input image size. In Figure 5b, we compare the performance of the three methods with the same relocalisaton feature map size of $200 \times 100$. DualRC-Net achieves the best performance under huge illumination variations, but it does not perform as well as the other two methods under huge viewpoint variations. Overall, DualRC-Net performs on par with both Sparse-NCNet and NCNet. We hypothesise two reasons for the deterioration in huge viewpoint variation. One is that the input image size for DualRC-Net is only a half of Sparse-NCNet and a quarter of NCNet hence some valuable information are lost in the down-sampled input image. Second, the 4D correlation tensor of DualRC-Net is only $50 \times 38 \times 50 \times 38$ while the other two methods have the size of $100 \times 75 \times 100 \times 75$. Smaller 4D correlation tensor indicates weaker filtering effect from the neighbourhood consensus module which could lead to worse performance under huge viewpoint changes. In Figure 5c we demonstrate that DualRC-Net can achieve similar or better results on HPatches with a much smaller image size than other methods, which further verifies the effectiveness of our dual-resolution design in DualRC-Net. We also evaluate the runtime and memory cost for each method. The average processing time per image pair are 2.05s/0.82s/4.15s by DualRC-Net/Sparse-NCNet/NCNet with GPU memory of 1232MB/680MB/7868MB respectively on a $200 \times 150$ feature map. The evaluation is performed on Nvidia GTX 1080Ti.

We notice an interesting trend in Figure 5 that a larger relocalisation resolution leads to better performance under illumination changes but worse performance under viewpoint changes at small thresholds. This could be explained as follows. A smaller localisation resolution means that features are more sparsely distributed on the image. Each feature covers a greater image area so that the difference between nearby features is more distinctive. Moreover, as there is no homographic change between the image pair, the correct correspondence of one point in image $\mathbf{I}^A$ will be at the same location in image $\mathbf{I}^B$, hence there is no need to search other places. Therefore, feature resolution has little impact on the performance under illumination change. However, under viewpoint changes, the relocalisation feature resolution matters more. If the resolution is small, the model will not be able to capture the viewpoint changes as the features are spatially too coarse compared with the original image. With a larger feature resolution, the discrepancy between nearby locations can be properly captured, therefore, more robust matching can be achieved.

(a) All methods have the same input image size of $1600 \times 1200$. The relocalisation feature resolutions of DualRC-Net, Sparse-NCNet and NC-Net are $400 \times 300$, $200 \times 100$ and $100 \times 75$ respectively.

(b) All methods have the same relocalisation resolution of $200 \times 150$. The input image sizes for DualRC-Net, Sparse-NCNet, and NCNet are $800 \times 600$, $1600 \times 1200$ and $3200 \times 2400$ respectively.

(c) DualRC-Net can achieve better results with a smaller input image size. The input image sizes for DualRC-Net, Sparse-NCNet and NCNet are $1200 \times 900$, $1600 \times 1200$ and $3200 \times 2400$ respectively.

Figure 5: Comparing DualRC-Net with Sparse-NCNet and NCNet.

# 3 Qualitative comparison

In this section, we provide more qualitative results of DualRC-Net on HPatches, InLoc, and Aachen Day-Night benchmarks and compare with Sparse-NCNet and NCNet. The results for Sparse-NCNet and NCNet are obtained using the pre-trained models provided by the authors under the best configurations reported in the original papers. The input image sizes are $1600 \times 1200$ for DualRC-Net and $3200 \times 2400$ for both Sparse-NCNet and NCNet.

**HPatches benchmark**  We qualitatively compare DualRC-Net with Sparse-NCNet and NCNet on HPatches benchmark [1]. Top 6000 matches are selected for each method. The correspondences with a re-projection error less than 3 pixels are considered to be correct and are marked as green dots. Otherwise, the correspondences are considered to be wrong and are marked as red dots. We demonstrate that DualRC-Net performs notably better than others under challenging illumination variations and performs on par with others under huge viewpoint changes.

| DualRC-Net | Sparse-NCNet | NCNet |
|:---:|:---:|:---:|

| (4778/6000) | (985/6000) | (892/6000) |
|:---:|:---:|:---:|
| (5885/6000) | (4249/6000) | (4133/6000) |

Figure 6: Comparison on HPatches benchmark under huge illumination changes. Top 6000 matches are selected and the threshold is set as 3 pixels. DualRC-Net performs the best under challenging illumination variations. We also report (correct/total) matches under each pair.

DualRC-Net        Sparse-NCNet        NCNet

(4632/6000)        (4510/6000)        (2553/6000)

(5098/6000)        (5625/6000)        (2455/6000)

Figure 7: Comparison on HPatches benchmark under huge viewpoint changes. DualRC-Net substantially outperforms NCNet and performs on par with Sparse-NCNet. We also report (correct/total) matches under each pair.

**InLoc benchmark** We qualitatively compare DualRC-Net with Sparse-NCNet on InLoc benchmark [2]. Query images were taken a few months after the database images being collected. As no ground truth is provided, we can only assess the quality of correspondences by visual inspection. In Figure 8, we demonstrate that DualRC-Net is able to establish reliable correspondences for indoor scenes having very large viewpoint variations, illumination changes and repetitive patterns.

| DualRC-Net | Sparse-NCNet | DualRC-Net | Sparse-NCNet |
|---|---|---|---|

Figure 8: Qualitative comparison on InLoc. Top 500 matches are selected for each method.

**Aachen Day-Night benchmark** We qualitatively compare DualRC-Net with Sparse-NCNet on Aachen Day-Night benchmark [6]. As no ground truth is available, we can only assess the quality of the correspondence by visual inspection. From Figure 9, DualRC-Net is able to produce reliable correspondences for outdoor scenes having very large viewpoint changes, day-night illumination changes and repetitive patterns.

| DualRC-Net | Sparse-NCNet | DualRC-Net | Sparse-NCNet |
|---|---|---|---|

Figure 9: Qualitative comparison on Aachen Day-Night. Top 500 matches are selected for each method.