[Reviews · NeurIPS 2020]

Review 1

Summary and Contributions: This work introduces a novel neural network architecture, DRC-Net, for dense pixel-wise geometric matching of image pairs. DRC-Net first processes and low-resolution correlation tensor for coarse but robust correspondences. It is then combined with a local, high-resolution correlation tensor, yielding more precise matches while retaining the robustness. The paper provides evaluation on planar scenes, and indoor and outdoor 6-DoF localization.

Strengths: 1. The method is technically sound and a clever and novel extension of NCNet. Given prior works, it is relatively simple (this is a strength!) but nonetheless effective. Adding fine scores and upsampled coarse scores has been pretty popular recently: for keypoint detection in [A] and sparse-to-dense matching in [B]. 2. The evaluation is extensive and gives a good idea of the capability of the proposed approach. It is based on popular benchmarks that are sound and reliable. There are concerns of missing baselines and inappropriate metrics (see below), but the evaluation is already solid. [A] Zixin Luo, Lei Zhou, Xuyang Bai, Hongkai Chen, Jiahui Zhang, Yao Yao, Shiwei Li, Tian Fang, and Long Quan. ASLFeat: Learning Local Features of Accurate Shape and Localization. CVPR 2020. [B] Hugo Germain, Guillaume Bourmaud, and Vincent Lepetit. S2DNet: Learning Accurate Correspondences for Sparse-to-Dense Feature Matching. ECCV 2020.

Weaknesses: Although the evaluation datasets are sound and reliable, I am concerned with overclaiming. The paper claims at multiple occasions to achieve state-of-the-art results (lines 14, 63, 273) or even substantially outperform them (line 199). These claims seem reasonable given the reported results, but I think that they are actually not valid: 1. The baselines that are compared against are mostly other dense matching networks (Sparse-/NCNet in Figures 3, 6, and 7) or learned local features (D2Net, R2D2, SuperPoint in Figures 5 and 7). The evaluation does not include deep neural networks for sparse feature matching, which, similarly to DRC-Net, leverage information from both images, and are already referenced in the related work (line 76). i. On the Aachen Day-Night dataset, [the CVPR 2020 visual localization challenge](https://www.visuallocalization.net/workshop/cvpr/2020/) reports that *SuperPoint + SuperGlue* [C] achieves 45.9 / 70.4 / 88.8 and *ASLFeat (10k) + OANet* [A, D] achieves 48.0 / 67.3 / 88.8. These are notably higher than the results reported for DRC-Net, and thus invalidate the claim of state-of-the-art. (I also note that the authors should also report the updated results with more accurate ground truth, as it is now the default on the evaluation server). ii. On the HPatches dataset, ASLFeat reports impressive numbers (Figure 4 in the paper) and has publicly available code. It would also be possible to report SuperPoint + SuperGlue, given that the code is also public. iii. On the InLoc dataset, there are public entries for SuperPoint + SuperGlue, but as actual numbers and not cumulative plots reported here. Maybe including a table for InLoc would ease the comparison. 2. I suspect that the metric reported for HPatches, the mean matching accuracy (MMA), unfairly favors learned matchers, since it evaluate the precision, which is easy to maximize when the network can see both images at the same time (DRC-Net) but hard when it seems only one image at a time (local features + nearest neighbor matching). It is only fair if all methods have exactly the same number of matches, which seems to be only approximately enforced according to the caption of Figure 5. The authors should either strictly enforce it, or rely on higher level metrics like homography estimation, as reported in SuperPoint. Additionally, given the small size of the Aachen Day-Night dataset, evaluating on an additional outdoor dataset would be highly valuable. There are several, not mutually-exclusive, options: - Run the evaluation on the second version of the dataset, Aachen v1.1 [E], which is larger. - Run the full localization on both day and night queries, e.g. using the [hloc toolbox](https://github.com/cvg/Hierarchical-Localization/). - Reports results on other datasets, like the [CVPR 2020 Image Matching Challenge](https://vision.uvic.ca/image-matching-challenge/). [C] Paul-Edouard Sarlin, Daniel DeTone, Tomasz Malisiewicz, and Andrew Rabinovich. SuperGlue: Learning Feature Matching with Graph Neural Networks. CVPR 2020. [D] Jiahui Zhang, Dawei Sun, Zixin Luo, Anbang Yao, Lei Zhou, Tianwei Shen, Yurong Chen, Long Quan, and Hongen Liao. Learning Two-View Correspondences and Geometry Using Order-Aware Network. ICCV 2019. [E] Zichao Zhang, Torsten Sattler, and Davide Scaramuzza. Reference Pose Generation for Visual Localization via Learned Features and View Synthesis. Arxiv 2005.05179.

Correctness: As explained previously, the claims are correct given the reported numbers, but I believe that some baselines are missing.

Clarity: The paper is very well-written - the writing is very clear and pleasant to read. The figures are beautiful and illustrate very well the mains steps of the algorithm. The mathematical notations are well-chosen and clear. I also appreciate tha the authors managed to fit qualitative results in the main paper. Great job! The broader impact statement is very nicely written - I agree with the balanced and optimistic conclusion.

Relation to Prior Work: The related work is well written. I think that mentioning the similarity to ASLFeat [A] and S2DNet [B] in terms or coarse-to-fine paradigm would be valuable to the community.

Reproducibility: Yes

Additional Feedback: I appreciate that the authors provided code and demo with a proper documentation. The code was not always clear and readable, but it is better than no code. I suggest to address the issues mentioned in the weakness section. I am willing to increase my rating if the rebutal adresses some these concerns. The paper is already great, but that would make it 100% sound and fair. I have additional suggestions and questions regarding the method: - The soft mutual nearest neighbor, introduced in NCNet, seems to sometimes produce unexpected effects. Since the input $\tilde{C}_{ijkl}$ can take negative values (in [-1, 1] for the first block and unconstrained for the second block), dividing by the maximum can flip the sign, and actually increase weak scores. Considering that the correlation is instead a log-probability, one should actually substract the maximum, e.g. $\bar{C}_{ijkl} = 3\tilde{C}_{ijkl} - \text{max}_{ab} \tilde{C}_{abkl} - \text{max}_{cd} \tilde{C}_{ijcd}$. To improve the gradient flow, one could also use a soft maximum with log-sum-exp (LSE): $\bar{C}_{ijkl} = 3\tilde{C}_{ijkl} - \text{LSE}_{ab} \tilde{C}_{abkl} - \text{LSE}_{cd} \tilde{C}_{ijcd}$. This is very similar to a single step of the Sinkhorn algorithm (as used in [C]), which does this normalization alternatively. - I recall that NCNet normalizes the output correlation with a softmax. In Section 3.3 it seems that both $\hat{S}$ and $\tilde{S}$ are unnormalized, and can then take negative values. I think that this can cause an issue in Equation 3, where two weak negative scores could become positive after multiplication. I looked at the code but found the softmax commented out. - Lastly, I think that computing the upsampled locations as $i = i'r=i'\frac{H_a}{h_a}$ is only valid if the downsampling of the features in the backbone is done with stride-2 convolutions. If it is done with stride-2 max pooling (which is the case once in ResNet-101), the downsampled location is at the center of the 2x2 patch (and not at the top left corner). For a x2 downsampling, the new location is: $i' = \frac{i}{2} - \frac{1}{4}$. Using this I saw an improvement for feature localization, so it should help in your case too - sadly I don't have any reference for this implementation detail. ========== Post-rebuttal update: I have read the rebuttal and appreciate that the authors added comparisons to additional baselines and updated the numbers on the Aachen Day-Night dataset. The results there look great and do not need to beat the current state of the art to be worthy of publication. I also appreciate the clarifications regarding the use of the softmax, runtime, and memory requirements. I however still share the concerns of other reviewers regarding the inappropriate metrics in HPatches, and the lack of clarity of the method description. I will therefore not increase my rating. I am also still not satisfied with the reporting of the InLoc results, as the [benchmark website](https://www.visuallocalization.net/benchmark/) reports separate numbers for buildings DUC1 and DUC2, which are not comparable to the table provided in the rebuttal.


Review 2

Summary and Contributions: The paper presents a variant of NCNet (trained with principles from Sparse NCNet) where matching is done at two resolutions, from coarse to fine. This reduces the memory footprint and allows for the use of larger images. Results are hard to parse, despite the SOTA claims, and it is hard to differentiate the method from its predecessors.

Strengths: - Well-written paper (with some caveats). - Good concept, alleviating the main issue of NCNet (requires dense correlation maps with a big memory footprint).

Weaknesses: - Small contributions over previous methods (NCNet [6] and Sparse NCNet [21]). Mostly (good) engineering. And despite that it seems hard to differentiate it from its predecessors, as it performs very similarly in practice. - Claims to be SOTA on three datasets, but this does not seem to be the case. Does not evaluate on what it trains on (see "additional feedback").

Correctness: No issues regarding the method. I have moved my comments regarding the experiments to "additional feedback" since this field is limited to 2k characters from some reason. L154-156: "we increase the number of output channels of the 1x1 conv kernels from 256 to that of the highest feature map (i.e. 1024). The aim is to increase the complexity of the descriptor and hence achieve higher accuracy during matching." How does this help? It seems reasonable (although that's a very high number for a local descriptor) as a way to combine layers with different dimensionalities without losing information, but you do not gain anything else from it.

Clarity: L167: "Notice that (i', j') may not be integers and hence we apply bilinear interpolation [...]". I don't follow this. If (i, j) are coarse coordinates and (i', j') are refined coordinates, and the resolution of the fine feature maps is 4x that of the coarse feature maps, both should be integers? I think this might just be bad phrasing, or I may be missing something. L173: How is multiplying two score matrices equivalent to applying a mask? Eqs. 1 and 2: shouldn't the left-hand side not have the jk sub-indices? The notation is confusing.

Relation to Prior Work: L17 "3D reconstruction [1, 2, 3]". I'm not sure what OANet ([3]) is doing on that list? It is an outlier rejector for pairwise matching, which is just an early step in SfM.

Reproducibility: Yes

Additional Feedback: I have several notes regarding the experiments. (1) On HPatches I don't understand why the MMA evaluation on HPatches has become a standard benchmark for local features. You could very well chop it off at 3 pixels: why does it matter how well a feature matches 10 pixels away? That is extremely inaccurate. And feature matching does not measure performance on homography estimation (which does not need many points to begin with). Fig. 5 caption: I don't understand how it's fair to select 2k points for the baseline methods when the proposed method is doing dense feature map correlation. D2-Net and R2D2 work significantly better with a larger budget, and those are the results they report in their respective papers. This decision seems arbitrary. And the paper could use more baselines anyway (I suspect that learned patch descriptors on SIFT keypoints would work very well here). (2) On InLoc Performance is essentially identical as the two closest baselines (NCNet and sparse NCNet), despite the (presumed) difference in resolution. As for HPatches, results at large error thresholds are not specially significant. Moreover, the InLoc datasets only consider the original InLoc paper baselines and NCNet and Sparse NCNet, i.e. directly related methods. Unfortunately I don't know how to compare the results, but the visual localization challenge (visuallocalization.net) includes this dataset since 2019 and NCNet does not seem to be SOTA on it. (3) On Aachen Same for Aachen: while more baselines are considered, results are essentially identical to Sparse NCNet (vanilla NCNet is not listed); see Fig. 7 for details. And the SOTA on this dataset is at 86.7 / 93.9 / 100.0 (visuallocalization.net/benchmark) which is quite far from 44.9 / 68.4 / 88.8. This is including entries with learned matchers, but I think this is a fair comparison for a heavy, dense, correlation-based method such as this. --- On a separate note, I find it quite strange that the method is trained on Megadepth (outdoors, wide baselines) and evaluated on (1) image pairs with nothing but homographies or illumination changes (but not both), (2) an indoors dataset, and (3) and extreme day-to-night dataset. None of these scenarios are well-represented in Megadepth, as far as I know. If the method is working well on challenging wide-baseline matching of urban scenes it should be in the authors' interest to evaluate it on that scenario, which is far from solved. Results on a test set of the MegaDepth data, with good baselines, would be quite informative already, but in addition to that, multiple papers evaluate on YFCC now (vision.uvic.ca/image-matching-challenge), and the visual localization challenge (visuallocalization.net) contains not only Aachen and InLoc (in a more interpretable/comparable form than what is presented in the paper) but also more relevant datasets such as RobotCar or CMU Seasons. Other comments: L62: "large-scale public benchmarks including HPatches". Hpatches provides a nice data point, but it is not "large scale". Fig. 1 would be easier to parse if the left-side "bubbles" were reversed. Fig. 6: why are the data points on such odd coordinates? For InLoc errors are typically reported at 0.25m, 0.5m, etc. Typos, etc: L14: "It achieves state-of-the-art results" L16: "share" Fig. 1 caption: selected -> select L145: "4D convolutions-based" Fig 2: I don't understand what the layer names mean, but I guess "conv4_23" is a typo. L245: DELF not DEFL ------ Upgraded my rating since I understand now why some scores were not compatible with the previous version of the dataset. It would have been helpful if the authors explained this. Still unsure how the method performs on real settings.


Review 3

Summary and Contributions: This paper proposes a deep learning solution to search correspondence. The core idea is to build feature map in two resolutions, one in low res and the other in full res, and build 4D correlation volume on each. Then the computational expensive 4D convolution needs to be run only on the low-res one, which produces a guide to mask out invalid correspondence in high-res 4D volume. The method works on par or better than SOTA on several popular benchmarks.

Strengths: - The idea is technically sound. - The method is evaluated on several popular and challenging benchmarks. - Qualitative evaluation is also given in supp, which I think would be better to place in the main paper, as it is important to understand the advantages of the method.

Weaknesses: - The novelty seems limited. The idea of building correlation in low-res and then refine for or facilitate the high-res results might be new in the literature for correspondence search, but it is quite common and has been widely adopted in previous work doing stereo match, where the main job is also to find correspondence but along epipolar line. - The main contribution of this paper, IMO, is to run 4D convolution on low-res correlation volume, which saves computation and possibly achieve comparable performance. If so, the experiment showing the saving of computational resources, e.g. gpu runtime, flip-flop, memory, must be given. - Similar multi-scale approach in stereo matching often runs fast at the cost of losing accuracy, since the correlation volume in low-res is not as informative as the high-res one, and it is not easy to fix if some mistakes are made in low-res. However the experiments show that the result is even better than SOTA. It would be good to add more explanation and analysis. - It would be nice and inspiring to show some qualitative results, possibly with zoomed-in view, for cases where previous methods failed but okay with the proposed method. Also, it's good to show some failure cases and analysis the limitation.

Correctness: Yes.

Clarity: Yes.

Relation to Prior Work: Yes, though I recommend adding some references about stereo matching since similar multi-scale idea has been widely used there.

Reproducibility: Yes

Additional Feedback: I like the multi-scale idea, but think the experiment section should be more focus and supportive for the main contribution.


Review 4

Summary and Contributions: The paper addresses the problem of establishing dense pixel-wise correspondence between a pair of image, which is an active area of research. The author(s) build on top of NC-Net framework, by proposing an FPN-like layer, to avoid using the full resolution maps for filtering with the 4D convolution.

Strengths: - novel idea, using FPN-like backbone to achieve coarse to fine matching; - reduced the number of inputs to the bottleneck of the algorithm, i.e. 4D convolution; - thorough empirical evaluation on Aachen day-night; Inloc and HPatches;

Weaknesses: - limited comparison with other, non NC-Net based methods: Yi et al, Learning to find good correspondences; OA-Net of Zhang et al (Learning Two-View Correspondences and Geometry Using Order-Aware Network) MegaDepth dataset contains outdoor images; did the authors consider training on (the smaller) IVD dataset, provided with the NC-Net paper? Comparison with Sparse NC-Net / NC-Net could be moved in the main paper.

Correctness: seems correct;

Clarity: Sec. 3.3 could be improved, by having an intro sentence explaining what follows, i.e. "masking" the finer matches with the coarse matches coming from interpolated C~. It might be easier to read / more intuitive if the indices denoted with ', i.e. i', j' were in the coarse resolution (but no strong opinion on that).

Relation to Prior Work: Compares to recent works, and analysis with NC-Nets (Sparse and "original") is detailed; the comparison could be moved to the main paper. Some other non NC-Net methods could be referenced.

Reproducibility: Yes

Additional Feedback: Shared the concerns of the other reviewers in terms of clarity of the method and evaluation. However, it is still a good submission. The points from the reviews were addressed in the rebuttal, and will be updated in the final version. Updated the score to 7. --------------- - would be great to add some intuition why DRC-Net seems to perform better on large variation in illumination; There is no comparison how non-isotropic filters of [21] could impact the proposed method.


Review 5

Summary and Contributions: The paper presents a method for obtaining correspondences between pairs of images, which extends the NCNet method [5] by adding a two-stage approach which operates on two different resolutions. This allows to obtain more accurately localized correspondences while avoiding the need to run NCNet in high-res.

Strengths: - One of the main limitations of the NCNet work is its computational cost. This paper proposes a way to address this limitation, by running the 4D match filtering NCNet at a small feature resolution, and then refining the matches using the higher resolution features. - The paper obtains good experimental results. In particular, qualitative results on Aachen and Inloc on the supp. mat. show an improvement over the recent Sparse-NCNet.

Weaknesses: - Section 3 could benefit from some reorganization. I think it would be beneficial to present the "Dual-resolution feature map generation" section before the "Neighbourhood consensus filtering" as this is the order in which it occurs during excecution. - In line with the previous point, a small section "Selection of query points" should be added before "Masking and dense matching", as from Fig. 1 I understand that the high-resolution matching is only done on a subset of query points. - No details are given about the architecture used for the 4D NCNet, in terms of number of conv. layers and kernel sizes, etc. - The explanation about Sparse-NCNet in L92-94 seems incorrect. These are actually 4D convolutions, but which act on the active sites only (thus the name submanifold). - While the authors don't claim to produce a more efficient method, it would be good to know the memory requirements and execution times of their method, compared to the other Neighbourhood Consensus methods.

Correctness: Yes.

Clarity: Some improvements could be made in terms of organization as noted in the "weaknesses" answer, as well as in terms of the explanation of the query selection and high-resolution matching.

Relation to Prior Work: Yes.

Reproducibility: Yes

Additional Feedback: After seeing the rebuttal and the other reviews, I have increased my score from 6 to 7 as the method obtained SOTA results at submission and the additional results from the rebuttal show that the method is robust to strong illumination changes, beating strong a baseline based on local features (SuperPoint+SuperGlue). (As a side note: Unfortunately HPatches does not feature a "Illumination & Viewpoint" test set, which I think should be created by synthetically warping the illumination sequences using the homographies from the viewpoint sequences, as it would give additional insights of the performance under combined viewpoint and illumination changes, which is closer to reality.)

[Author Response · NeurIPS 2020]

We appreciate all reviewers for their valuable comments and confirming the simplicity of our design and the repeatability
of our experiments. We address the main concerns below.

**R1,R2,R3: SuperGlue & OANet** First of all, the Aachen Day-Night updated its ground truth after our submission. We
now re-evaluate our method against all baselines and include the SuperPoint+SuperGlue (SP+SG) and ASLFeat+OANet
(upper left table). We also compared with SP+SG on HPatches using the standard MMA metric (the bottom right
figures) and the overall MMA of DRC-Net is significantly higher. On Aachen, DRC-Net performs comparably well
with SP+SG. By adding Orthogonal Loss [21], the accuracy under low error threshold is improved. We believe injecting
SuperPoint into our framework will further boost DRC-Net. Note that SP+SG and ASLFeat+OANet were published in
CVPR20 (after our submission). DRC-Net was SOTA during submission. **R1,R2: Comparison on InLoc in table**
The plots of InLoc intended to emphasise the robustness and stability of all methods, and we also provide a comparison
in table (lower left table). **R1,R2: MMA for fair comparison** We believe it is fair because we follow the identical
evaluation protocol as [5,6]. As described in [6], mutual NN is applied on other description-matching methods to obtain
about 1k matches to ensure a comparable number of matches. Therefore, MMA is evaluated on nearly an identical
number of matches for fair comparison. **R1,R2: Experiments on more datasets** We follow the suggestion to evaluate
on Aachen v1.1 and the results are $71.2/86.9/97.9$. We will include more results in the final version. **R1: Negative**
**scores and softmax** The ReLU layers are employed in neighbourhood consensus module, hence it is guaranteed the
output scores at the coarse level are non-negative, thus adding softmax becomes optional. We choose to switch off
softmax as we found softmax slows down the training convergence, possibly because of reduced gradient after softmax.
**R2,R3: Novelty** DRC-Net is inspired by NCNet but significantly different. DRC-Net tackles the scalability issue
of dense matching with the subtle design of dual-resolution feature framework, which can effectively make use of
feature maps of different resolutions, substantially outperforming all neighbourhood consensus based methods. **R2:**
**Same training principle as Sparse-NCNet** Our training principle is different from Sparse-NCNet. Sparse-NCNet is
supervised by image level annotations, while DRC-Net is supervised by sparse keypoint annotations. The training
losses are different as well, which will be clarified in the final version. **R2: Why 1024 channels** The use of 1024
channels is inherited from [5,6]. We also find that using 256 channels in our model can provide comparable ( slightly
inferior) accuracy which has been reported in Fig. 4 in supplementary. **R2: Insignificance of reporting performance**
**over large error band** The performance over large error band represents the stability and robustness. It is a common
practice to plot up to 2m for InLoc [24,5,6] and 10 pixel for HPatches [6,7,8]. This is meaningful because the relative
errors of 10 pixel are about 1% in HPatches and less than 10% at 1m for InLoc. Our method is superior than baselines
in these circumstances. See lower left table for details. **R2,R4: Notation and "mask"** $(i', j')$ is for coarse-level
and $(i, j)$ is for fine-level coordinates. We will further clarify. We use "mask" to indicate that some fine-resolution
scores would be zeroed by coarse-resolution scores since the ReLU layers enable zeros in 4D tensor. This can be an
analogue to binary mask. We follow the notation convention used in [5,6,21] to use $ijkl$ to index a correlation score.
**R2,R4: Why train on MegaDepth, not on IVD** Training on MegaDepth and testing on the localisation datasets for
establishing correspondence has been successfully adopted in literatures (See [7,8,40] and S2DNet by Germain et al
2020) for MegaDepth contains rich viewpoint and illumination variations. Testing on other datasets with standard
3D reconstruction pipelines allows fair comparison with a large number of baselines. IVD is a popular alternative,
however, IVD lacks of the sparse pixel-wise annotation required to train DRC-Net. **R3,R5: Runtime and memory**
We follow the suggestion and evaluate the runtime/GPU memory on a fixed feature map size $200 \times 150$ for three
methods, the average processing time per image pair are 2.05s/0.82s/4.15s by DRC-Net/Sparse-NCNet/NCNet with
GPU memory cost of 1232MB/680MB/7868MB respectively. All three methods are evaluated on a GTX 1080Ti GPU.
**R3,R4: Performance in illumination** Please refer to sect. B supplementary. It will be included in the main paper in
the final version. **R3: Qualitative results and failure** Please refer to sect. C in supp. Failure cases will be included and
discussed in the final version. **R4: Including non-isotropic filters** We have tried to include similar adaptive module
as [21] into our framework, but no obvious gain is observed possibly because the only feasible non-isotropic filters is
small and hence inadequate to deal with strong perspective scale variation. **R5: NC module configuration** We use the
same configuration as NCNet (as mentioned in line 206). It will be further clarified. **R1,R2,R5: Typos and figures**
Thanks, all will be fixed.

| Method | Correctly localized queries (%) | | |
| --- | --- | --- | --- |
| | 0.25m, 2° | 0.5m, 5° | 5m, 10° |
| SP+SG | **79.6** | **90.8** | 100.0 |
| Sparse-NCNet | 76.5 | 84.7 | _98.0_ |
| ASLFeat+OANet | 77.6 | _89.8_ | 100.0 |
| DRC-Net w/o orth. | _78.6_ | 88.8 | 100.0 |
| DRC-Net w/ orth. | **79.6** | 88.8 | 100.0 |

| Method | Correctly localized queries (%) | | |
| --- | --- | --- | --- |
| | 0.25m | 0.5m | 1m |
| IL+DRC-Net | 43.2 | _67.5_ | **83.0** |
| IL+DRC-Net (Type(e)) | **48.0** | **68.1** | **83.0** |
| IL+D2-Net | 43.2 | 61.1 | 74.2 |
| IL+Sparse-NCNet | _45.4_ | 66.2 | _79.9_ |

Upper left table: Aachen Day-Night. Lower left table: InLoc. Three figs on the right: DRC-Net vs SP+SG on HPatches.

[Meta-Review · NeurIPS 2020]

The initial scores for this paper were diverging: 3: A clear reject. 6: Marginally above the acceptance threshold. 8: Top 50% of accepted NeurIPS papers. A very good submission; a clear accept. 7: A good submission; accept. 5: Marginally below the acceptance threshold. The positive points praised by the reviewers were: - Technically sound, clever, novel, simple and effective method. - Extensive evaluation on popular benchmarks that are sound and reliable. - Very well written paper. - Nice qualitative evaluation. The negative points: - Concerns regarding missing comparisons with important baselines and choice of metrics. - Limited novelty. - Missing run-time comparisons. - Missing analysis why the results are good. - Presentation: Some sections could be re-ordered, some missing details. The authors provide a rebuttal, which addresses many of the weak points. After reading the rebuttal and in the post-rebuttal discussion, R5 increases their score from 6 to 7, R1 maintains their positive rating 7, R4 still likes the paper but lowers their score from 8 to 7 taking into account comments from the other reviewers. In view of the rebuttal, the negative reviewer upgrades their score from 3 to 4 as the rebuttal clarifies some of the issues, but still has concerns “how the method performs on real settings.” The borderline negative R3 (5) did not participate in the discussion or provided a post-rebuttal comment. The final scores of the paper are: 7, 7, 7, 5, 4. The AC has looked at the reviews, rebuttal and has read the paper. The AC is convinced by the arguments of the three positive reviews and recommends Accept.